# A new perspective on transient characteristics of quiet stance postural control

**Cody A. Reed[1], Ajit M. W. Chaudhari [2], Lise C. Worthen-Chaudhari [3], Kimberly E. Bigelow[4], Scott M. Monfort[1]** *

**1** Department of Mechanical and Industrial Engineering, Montana State University, Bozeman, Montana, United States of America, **2** School of Health and Rehabilitation Sciences, The Ohio State University, Columbus, Ohio, United States of America, **3** Department of Physical Medicine and Rehabilitation, The Ohio State University, Columbus, Ohio, United States of America, **4** Department of Mechanical and Aerospace Engineering, University of Dayton, Dayton, Ohio, United States of America

* scott.monfort@montana.edu

**Data Availability Statement:** All postural control files of epoch estimates are available from the Dryad database (https://datadryad.org/stash/share/

## Abstract

Postural control provides insight into health concerns such as fall risk but remains relatively untapped as a vital sign of health. One understudied aspect of postural control involves transient responses within center of pressure (CoP) data to events such as vision occlusion. Such responses are masked by common whole-trial analyses. We hypothesized that the transient behavior of postural control would yield unique and clinically-relevant information for quiet stance compared to traditionally calculated whole-trial CoP estimates. Three experiments were conducted to test different aspects of this central hypothesis. To test whether transient, epoch-based characteristics of CoP estimates provide different information than traditional whole-trial estimates, we investigated correlations between these estimates for a population of young adults performing three 60-second trials of quiet stance with eyes closed. Next, to test if transient behavior is a result of sensory reweighting after eye closure, we compared transient characteristics between eyes closed and eyes open conditions. Finally, to test if there was an effect of age on transient behavior, we compared transient characteristics during eyes closed stance between populations of young and older adults. Negligible correlations were found between transient characteristics and whole-trial estimates (p>0.08), demonstrating limited overlap in information between them. Additionally, transient behavior was exaggerated during eyes closed stance relative to eyes open (p<0.044). Lastly, we found that transient characteristics were able to distinguish between younger and older adults, supporting their clinical relevance (p<0.029). An epoch-based approach captured unique and potentially clinically-relevant postural control information compared to whole-trial estimates. While longer trials may improve the reliability of whole-trial estimates, including a complementary assessment of the initial transient characteristics may provide a more comprehensive characterization of postural control.

Cmoos4w5D_
pjV2ZVH56f9yH68FYi1YoCGyG7VqvEjh4).

**Funding:** The authors received no specific funding for this work.

**Competing interests:** The authors have declared that no competing interests exist.

# 1. Introduction

Postural control has been widely studied to provide insight into various negative health outcomes such as falls, musculoskeletal injuries, and concussions [1–3]. Center of pressure (CoP) parameters are often used as the primary outcome variables to characterize postural control for clinical applications, such as predicting fall risk [3, 4]. While these CoP parameters are clinically-relevant, there is significant untapped potential in using these postural control measures as an indicator of health, particularly when investigating transient responses within CoP data.

Traditionally, CoP analyses result in whole-trial estimates of 30 seconds or longer (up to 2 minutes) [3, 5–7]. However, reports of nonstationarity and sampling duration effects on CoP-based variables of interest raise questions about what aspects of postural control are represented by whole-trial estimates [7]. While longer sampling durations, of 1–2 minutes, have been proposed to increase CoP parameter reliability [6], this approach masks transient postural behavior (i.e., an initial destabilized period followed by a transition to a more stable, quasi-steady state level) that might be relevant to human health. For instance, transient postural control behavior associated with sensory transition (i.e., vision obstruction) was reported to distinguish adults with diabetes from healthy counterparts [8]. Still, a dearth of research into transient postural responses limits the ability to utilize them clinically, necessitating further study.

Although transient behavior has previously been identified during quiet stance [8–11], little research has attempted to characterize aspects of the transient response and understand the clinical relevance associated with this behavior. In experiments where transient behavior has been observed, the transient response often occurs after a sensory transition, such as eye closure, which suggests that this behavior may be a result of sensory reweighting [8–11]. Additionally, few experiments have studied transient postural behavior in older adults despite their well-documented increased risk of falling [3, 12, 13]. Therefore, deeper investigation into the transient behavior of common CoP parameters may yield new and unique information that complements existing whole-trial estimate approaches.

Three experiments were conducted with the overall purpose to better understand the clinical utility of transient characteristics of postural control during quiet stance. Experiment 1 tested whether transient characteristics of CoP estimates provide different information than traditional whole-trial estimates, with the hypothesis being that transient characteristics of CoP estimates would not be associated with whole-trial CoP estimates. Experiment 2 tested if transient behavior is a result of sensory reweighting after eye closure, with the hypothesis that transient behavior would only exist during eyes closed stance and not eyes open stance. Experiment 3 tested whether transient behavior could distinguish between young and older adults, with the hypothesis that older adults would demonstrate more exaggerated transient behavior than young adults.

## 2. Experiment 1: Whole-trial estimates vs transient characteristics of postural control

### 2.1 Experiment 1 methods

**2.1.1. Participants.** Young adults (18–35 years old) were recruited from a 2017 American Society of Biomechanics (ASB) Quick Study (i.e. attendees of the 2017 ASB annual meeting in Boulder, CO, USA who volunteered to participate in a society-facilitated "quick research study") and the Bozeman, MT community. The data from these two groups were combined. Individuals with a known neurological impairment, a prior lower extremity joint replacement surgery, or a lower extremity injury within the three months prior to testing were excluded.

**2.1.2. Protocol.** Before testing, all participants provided written informed consent approved by the Ohio State University Biomedical Sciences Institutional Review Board (Protocol No. 2017H0149) or the Montana State University Institutional Review Board (Protocol No. SM042618). After providing informed consent, participants completed a testing session that analyzed their postural control performance during quiet, eyes closed (QEC) stance. All tests were completed in a single visit.

Each testing session consisted of three successful, 60-second QEC trials. Participants stood without shoes and positioned the medial borders of their feet 5 cm apart [14]. A researcher instructed each participant to stand as still as possible while looking forward and keeping their arms relaxed at their sides. Whenever they felt ready, participants then counted down aloud '3-2-1-GO', simultaneously closing their eyes as they said 'GO', which initiated the start of the 60-second trial. Prior to the collection of any official trials, participants performed a 10-second practice trial in which researchers confirmed that the participant understood the counting down and eye closure protocol. Between trials, participants were allowed a self-selected amount of rest. Any trial where a participant lost their balance was not used for analysis and an additional successful trial was then performed to obtain a total of three successful trials. During each trial, vertical force and moments about the x and y axes were recorded at 1000 Hz using a balance plate (BP5046; Bertec Corp.; Columbus, OH) and captured using custom software written in LabView (National Instruments; Austin, TX).

The resulting data were analyzed in MATLAB using custom scripts (Version 2018b; Math-Works, Inc.; Natick, MA). Medio-lateral and anterior-posterior CoP time series data were calculated from the force and moment data using Bertec guidelines. All data were $4^{th}$ order Butterworth lowpass filtered at 20 Hz [5, 14, 15]. Traditional, whole-trial estimates and epoch-based estimates were calculated for three commonly used CoP parameters: 95% confidence ellipse area (EA), medial-lateral mean velocity (MVEL_ml), and medial-lateral root-mean-squared excursion (RMS_ml) [5] (Fig 1A). These CoP parameters were selected because they have been linked to fall risk, supporting their clinical relevance [3, 16–18]. Increases in all CoP parameters were interpreted as worse balance (i.e. more sway). Whole-trial estimates were calculated for each CoP parameter using data from each entire 60-second trial and averaged across all three trials for each participant. In addition to the whole-trial estimates, epoch-based estimates of each CoP parameter were calculated by dividing each trial into twelve, 5-second epochs and calculating each CoP parameter for each epoch (Fig 1B). CoP data were demeaned with epoch-specific mean values, rather than a whole-trial mean, in order to isolate the sway within a given epoch.

Two transient characteristics of these epoch-based estimates were calculated on their original scales to quantify features of the transient behavior of each CoP parameter (Fig 1B): the difference between the $1^{st}$ and final ($12^{th}$) epochs (DIF_ovr) and the difference between the $1^{st}$ and $3^{rd}$ epochs (DIF_13). The DIF_ovr characteristic was used to quantify the difference from the beginning of the trial to the end of the trial, while the DIF_13 characteristic was used to isolate the transient behavior early in the trial (i.e., over the first 15 seconds) before postural control reached a quasi-steady-state. Each transient characteristic was calculated for EA, MVEL_ml, and RMS_ml parameters.

We also attempted to characterize the transient behavior with an exponential decay fit by using the exponent 'b', which we termed 'DECAY', from the exponential function equation with a constant offset ($y = a \cdot e^{bx} + c$). While the group data (i.e., averaged epoch values across all trials and participants for a given condition or group) were modeled very well with the proposed exponential fit, considerable variability in the appropriateness of the fit was observed for fitting data for individual participants for a given condition. Poor fits were most prominent when fitting epoch data from individual trials, but persisted in 0–30% of participants when

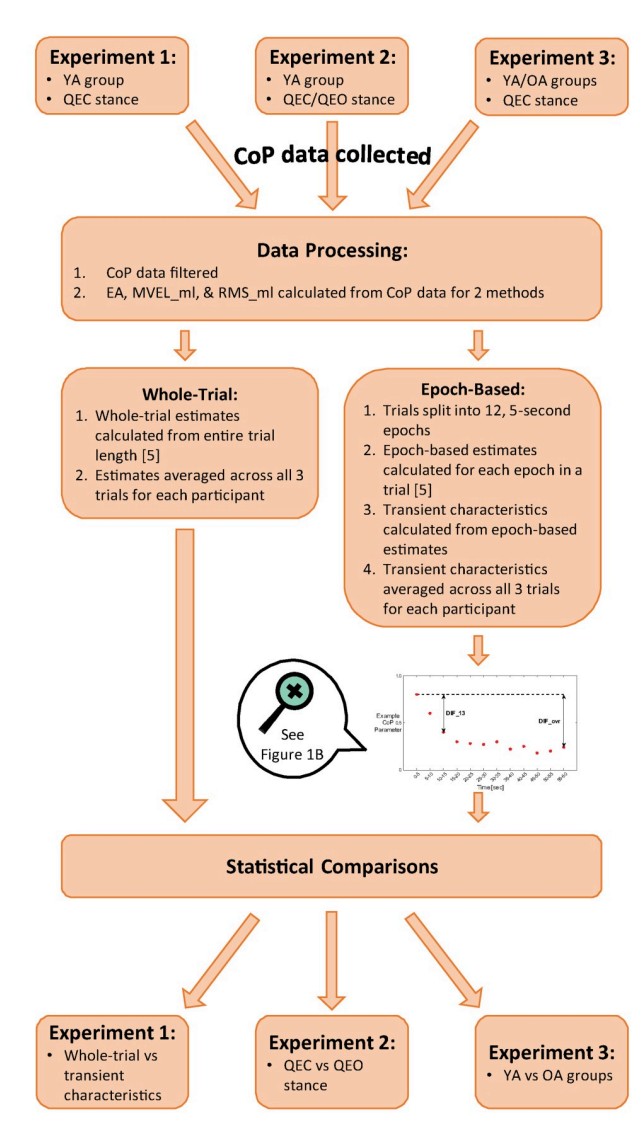

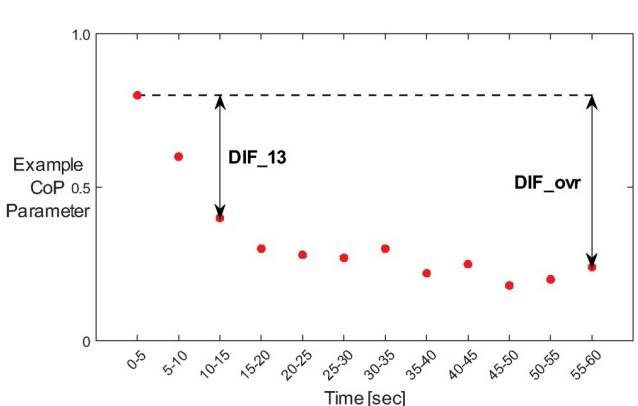

**Fig 1. Study overview.** (A) General study workflow. Abbreviation definitions: YA = young adult, OA = older adult, QEC = quiet eyes closed, QEO = quiet eyes open. (B) Magnified visual representation of epoch-based estimates and transient characteristics DIF_ovr and DIF_13 for a hypothetical response of an example CoP parameter (e.g., EA, MVEL_ml, or RMS_ml).

fitting epoch data that were averaged across trials for a given participant and condition. Cases of poor fits were highlighted by having negative $R^2$ values. Given the limitations of the poor fits, we elected to omit the DECAY analysis from the primary hypothesis testing for the three experiments, but have provided the associated analyses in the Supplemental content for reference (S1 Table). In these supplemental analyses, we account for the poor fits using three approaches: 1) including all DECAY values as calculated, 2) omit the data where the fits do not provide meaningful information (i.e., negative $R^2$), and 3) designate the DECAY values for these cases with a value of zero to indicate no observed transient behavior.

**2.1.3. Statistical analysis.** Spearman's Rank Order correlations were performed between the whole-trial estimates and the transient characteristics DIF_ovr and DIF_13. Transient characteristics DIF_ovr and DIF_13 were averaged across all three trials for each participant. Tests were run separately for EA-, MVEL_ml-, and RMS_ml-based transient characteristics. Non-parametric tests were used because data were often not normally distributed, as assessed by the Anderson-Darling normality test. Associations were interpreted using Spearman correlation coefficients on a traditional scale (.00 to .30 = *negligible*, .30 to .50 = *low*, .50 to .70 = *moderate*, .70 to .90 = *high*, .90 to 1.00 = *very high*) [19]. Significance for all analyses was defined *a priori* at α = 0.05. All analyses were performed in Minitab (Version 18.1; Minitab Inc., State College, PA).

## 2.2. Experiment 1 results

Sixty-seven healthy, young adults (24.9 ± 3.9 years, 75.7 ± 14.7 kg, 1.77 ± 0.09 m, 42 males/25 females) participated in the study. Transient characteristics DIF_ovr and DIF_13 were not correlated to the corresponding whole-trial estimates for EA, MVEL_ml, and RMS_ml parameters (all $P > 0.08$, Table 1). Group values of transient characteristics and whole-trial estimates are provided in Table 2 for reference.

# 3. Experiment 2: Eyes closed vs eyes open stance

## 3.1. Experiment 2 methods

**3.1.1. Participants.** For this experiment, a subset of 30 young adults (22.9 ± 2.6 years, 75.1 ± 11.6 kg, 1.77 ± 0.09 m, 18 males/12 females) that participated in Experiment 1 had additional data collected during their testing session which allowed us to look deeper into the effect of eye closure on transient postural behavior. These were the participants from the Bozeman, MT community that were not subjected to the same time constraints as the ASB Quick Study participants. The same exclusion criteria from Experiment 1 were also used in this experiment.

**Table 1. Spearman's rank-order correlations between transient characteristics and whole-trial estimates.**

|  | DIF_ovr | DIF_13 |
|---|---|---|
| EA | 0.01 (0.94) | 0.13 (0.31) |
| MVEL_ml | 0.21 (0.09) | 0.19 (0.12) |
| RMS_ml | -0.12 (0.32) | -0.02 (0.88) |

Values are presented as: Spearman's ρ (*P*-value)

**Table 2. Mean values for transient characteristics and whole-trial estimates for all CoP parameters.**

| CoP Parameter | Estimate | Value |
|---|---|---|
| EA [mm$^2$] | DIF_ovr | 86.0 ± 133.0 |
| | DIF_13 | 78.0 ± 129.6 |
| | Whole-Trial | 398.0 ± 263.7 |
| MVEL_ml [mm/s] | DIF_ovr | 4.72 ± 2.76 |
| | DIF_13 | 3.45 ± 2.99 |
| | Whole-Trial | 7.43 ± 2.60 |
| RMS_ml [mm] | DIF_ovr | 0.82 ± 1.03 |
| | DIF_13 | 0.62 ± 0.96 |
| | Whole-Trial | 3.99 ± 1.30 |

Mean ± SD values calculated for between-subjects.

**3.1.2. Protocol.** The subset of participants from Experiment 1 had their postural control performance analyzed during quiet, eyes open (QEO) stance in addition to QEC stance per the protocol approved by the Montana State University Institutional Review Board (Protocol No. SM042618). QEO tests were completed during the same visit as the QEC tests from Experiment 1.

The extended testing session consisted of three, 60-second QEC trials and three, 60-second QEO trials. The first trial was randomized between QEC and QEO conditions, with all subsequent trials alternating between the two conditions. For the three QEO trials, participants followed the same protocol described in Experiment 1 and counted down aloud '3-2-1-GO' as they did for QEC trials, but kept their eyes open and fixated on a target (fixation cross, 10 cm x 10 cm) placed 2 m away and 1.69 m high. Participants wore noise-canceling headphones for all QEC and QEO trials to minimize potential environment noise and audible distractions. All tests were completed in a single visit.

The same transient characteristics of epoch-based measures from Experiment 1 (i.e., DIF_ovr and DIF_13 for EA, MVEL_ml, and RMS_ml) were calculated for this experiment.

**3.1.3. Statistical analysis.** Linear mixed models were performed for QEC and QEO conditions, separately, to test for the effect of epoch on EA, MVEL_ml, and RMS_ml. Within the models, 'Participant' was a random effect, while 'Epoch', 'Trial Number', and 'Epoch*Trial Number' interaction were fixed effects. Normality and uniform distribution of model residuals were satisfied using the natural logarithms of all CoP parameters. Tukey post-hoc comparisons were performed with a family error rate of α = 0.05.

Additionally, paired t-tests were performed to test for the differences in transient characteristics between QEC and QEO conditions for all CoP parameters. Transient characteristics DIF_ovr and DIF_13 were averaged across all three trials for each participant. Because EA DIF_13 was not normally distributed, a 1-Sample Sign test of within-subject differences between conditions was performed for this variable instead of the paired t-test. Normality was assessed using the Anderson-Darling normality test. Significance for all analyses was defined *a priori* at α = 0.05. All analyses were performed in Minitab.

## 3.2. Experiment 2 results

The 'Epoch' fixed effect was significant for both QEC and QEO conditions (all $P < 0.018$) for EA, MVEL_ml, and RMS_ml parameters. Post-hoc analysis identified an initial transient period that was associated with worse balance, where the 1st and 2nd epochs (0–10 seconds) generally had the highest mean values for EA, MVEL_ml, and RMS_ml measures in both QEC

and QEO conditions (Fig 2), although not always statistically significant (S2 Table). Additionally, a significant effect of 'Trial Number' existed for all three CoP parameters in both QEC and QEO conditions (all $P < 0.018$), except EA during QEO ($P = 0.138$). Post-hoc analysis identified that in general, Trial 1 was associated with better balance compared to subsequent trials in both conditions, although not always statistically significant (S3 Table). No significant effect for the interaction 'Epoch*Trial Number' existed for any of the three CoP parameters for either QEC or QEO conditions (all $P > 0.059$).

Transient characteristics DIF_ovr and DIF_13 exhibited significant differences between QEC and QEO conditions for EA, MVEL_ml, and RMS_ml parameters, except for DIF_13 for RMS_ml (Table 3). The QEC condition consistently demonstrated higher DIF_ovr and DIF_13 estimates compared to the QEO condition, across all three CoP parameters (Table 3, Fig 2).

# 4. Experiment 3: Young vs older adults

## 4.1. Experiment 3 methods

**4.1.1. Participants.**   Older adults (OA) from the Bozeman, MT community were recruited for this experiment and had their postural control performance compared to the healthy, young adults (YA) from Experiment 1. Potential OA participants with any known neurological impairment or who could not stand for more than 5 minutes at a time without some form of assistance (e.g. cane, walker, etc.) were excluded.

**4.1.2. Protocol.**   Before testing, all participants provided written informed consent, which was approved by the Montana State University Institutional Review Board (Protocol No. SM042618). After providing informed consent, all participants completed a testing session that analyzed their postural control performance during QEC stance. All tests were completed in a single visit.

QEC trials were completed following the protocol described for Experiment 1 (i.e., 60-second trial, feet 5 cm apart, participant counted down '3-2-1-GO' and closed eyes to initiate trial). The same transient characteristics of epoch-based measures from Experiment 1 (i.e., DIF_ovr and DIF_13 for EA, MVEL_ml, and RMS_ml) were calculated for this experiment.

**4.1.3. Statistical analysis.**   To determine whether both young and older adult populations exhibited transient behavior in their postural control, linear mixed models were performed. This analysis was done for each population, separately, to test for the effect of epoch on EA, MVEL_ml, and RMS_ml. Within the models, 'Participant' was a random effect, while 'Epoch', 'Trial Number', and 'Epoch*Trial Number' were fixed effects. Normality and uniform distribution of model residuals were satisfied using the natural logarithms of all CoP parameters. Tukey post-hoc comparisons were performed with a family error rate of $\alpha = 0.05$.

Additionally, Kruskal-Wallis tests were performed to test for the differences in transient characteristics between OA and YA groups for each CoP parameter. Transient characteristics DIF_ovr and DIF_13 were averaged across all three trials for each participant. Non-parametric tests were used for these transient characteristic outcome measures because they were not normally distributed for most cases. Normality was assessed using the Anderson-Darling normality test. Significance for all analyses was defined *a priori* at $\alpha = 0.05$. All analyses were performed in Minitab.

## 4.2. Experiment 3 results

Sixty-seven healthy, young adults (YA, Table 4) and forty-nine older adults participated in the study. Eleven older adults were excluded due to the following reasons: 4, failure to comply with test protocol; 4, neurological impairment; 2, technical difficulties during collection; 1,

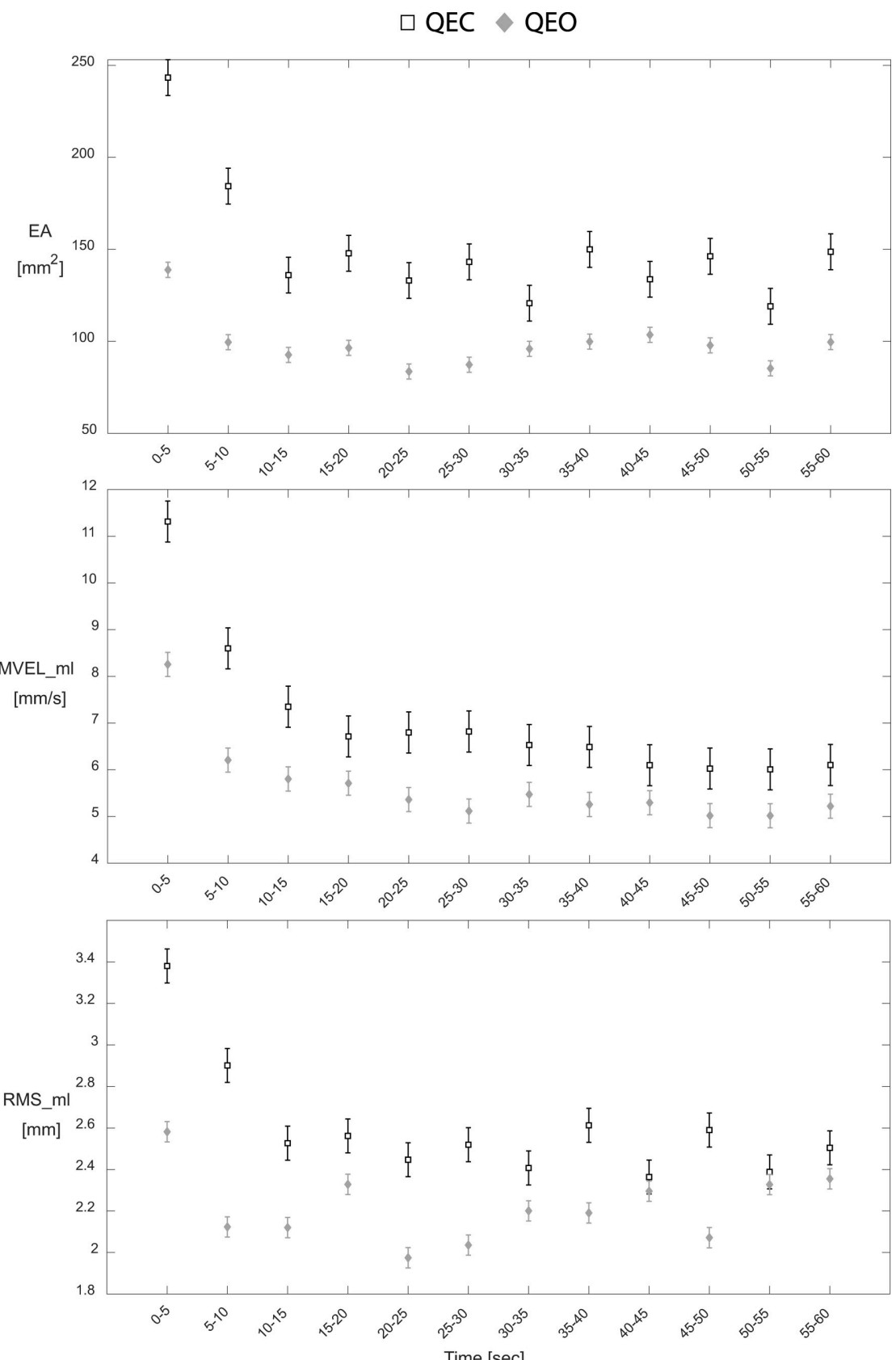

**Fig 2. QEC and QEO transient behavior for each CoP parameter.** Hollow squares and gray diamonds represent the time-series data for QEC stance and QEO stance, respectively. Values correspond to mean ± standard error for each epoch.

blind. The remaining 38 older adults (OA, Table 4) were included in the analyses for this experiment.

The 'Epoch' fixed effect was significant for both YA and OA groups (all $P < 0.013$) for all three CoP parameters, except RMS_ml for OA ($P = 0.105$). Post-hoc analysis identified an initial transient period that was associated with worse balance, where the 1st Epoch (0–5 seconds) generally had the highest mean values for EA, MVEL_ml, and RMS_ml measures in both YA and OA groups (Fig 3), although not always statistically significant (S4 Table). Additionally, a significant effect of 'Trial Number' existed for all three CoP parameters in both YA and OA groups (all $P < 0.006$), except EA for YA ($P = 0.058$). Post-hoc analysis identified that for OA, Trial 1 was generally associated with worse balance compared to subsequent trials, whereas for YA there was no common trend in how they performed from one trial to the next (S5 Table). No significant effect for the interaction 'Epoch*Trial Number' existed for any of the three CoP parameters in either age group (all $P > 0.225$).

Significant differences were found between the OA and YA groups for certain transient characteristics. The OA group had larger DIF_ovr for EA (Cohen's d = 0.71, $P = 0.001$) and DIF_ovr for RMS_ml (Cohen's d = 0.45, $P = 0.028$) (Table 5, Fig 3). No other transient characteristics exhibited significant differences between the two groups ($P > 0.05$, Table 5).

## 5. Discussion

This study represents a first step toward better understanding the clinical utility of transient characteristics of postural control during quiet stance. Our hypotheses were partially supported. Transient characteristics of epoch-based CoP estimates did not generally associate with whole-trial CoP estimates, supporting the limited overlap in the information they convey. Transient behavior (i.e., an initial destabilized period that precedes more stable postural control) was found for both eyes closed and eyes open conditions, although participants demonstrated exaggerated transient behavior consistent with greater deficits during the eyes closed condition. Additionally, older adults demonstrating more exaggerated transient behavior relative to young adults. This work supports the potential value of considering the transient responses in CoP data when assessing postural control. Notably, the analyses described for our experiments can be made with the same CoP time series data that researchers analyze when using traditional whole-trial estimates as primary outcomes. Even retrospective use of the

**Table 3. Mean values of transient characteristics for all CoP parameters for eyes closed and eyes open stance.**

| CoP Parameter | Transient Characteristic | QEC | QEO | *P*-value |
|---|---|---|---|---|
| EA [mm²] | DIF_ovr | 94.6 ± 126.2 | 39.3 ± 50.2 | **0.030*** |
| | DIF_13 | 107.3 ± 102.3 | 46.2 ± 60.8 | **0.043*†** |
| MVEL_ml [mm/s] | DIF_ovr | 5.21 ± 3.00 | 3.04 ± 1.60 | **0.001*** |
| | DIF_13 | 3.96 ± 2.86 | 2.45 ± 1.75 | **0.017*** |
| RMS_ml [mm] | DIF_ovr | 0.88 ± 0.94 | 0.23 ± 0.60 | **0.01*** |
| | DIF_13 | 0.85 ± 0.95 | 0.46 ± 0.74 | 0.118 |

* $P < 0.05$

† The *P*-value of DIF_13 for EA was obtained from a 1-Sample Sign test of within-subject differences between conditions because the data were not normally distributed

Mean ± SD values calculated for between-subjects.

**Table 4. Participant demographics (Mean ± SD).**

| Group | Young Adults (YA)[†] | Older Adults (OA) |
|---|---|---|
| Size (n) | 67 | 38 |
| Gender (m/f) | 42/25 | 8/30 |
| Age (years) | 24.9 ± 3.9[a] | 83.5 ± 8.4[a] |
| Mass (kg) | 75.7 ± 14.7[a] | 66.5 ± 12.9[a] |
| Height (m) | 1.77 ± 0.09[a] | 1.66 ± 0.09[a] |

[†] Young Adults group is the same as from Experiments 1 and 2. Demographics for this group are replicated from Section 2.2. for convenience.

[a] $P < 0.05$ for difference between YA and OA.

proposed transient measures can be feasible for protocols where balance trials were initiated simultaneously with a balance perturbation (sensor, cognitive, etc.) or if a trigger marked when the perturbation occurred. Therefore, the transient measures are accessible with little additional effort to provide a potentially more comprehensive assessment of postural control.

The current study found a lack of evidence for relationships between commonly-used whole trial estimates and transient characteristics of epoch-based estimates for the same type of outcome measure (e.g., EA, MVEL_ml, RMS_ml). These findings support the premise that unique information is contained in the initial transient responses of epoch-based CoP estimates that seems to be diminished when using CoP estimates based on entire trials. Conversely, a post-hoc analysis found that both the 1st and final (12th) epochs of EA, MVEL_ml, and RMS_ml measures exhibited moderate-to-high correlations with the whole-trial estimates (S6 Table). Therefore, while the epoch estimates from the beginning and end of trials may correlate to traditional whole-trial estimates, the characteristics of the transient behavior (e.g., differences between these epochs) reflect unique information (i.e., lack of correlation with whole-trial estimates). Additionally, when overlaying graphs of the whole-trial estimates and the time-series data of the same type of outcome measure (Fig 4), it can be seen that, especially for CoP parameters calculated from a central location (e.g., EA and RMS_ml), the whole-trial estimates deviate from the epoch-based estimates. While the whole-trial estimates of EA and RMS_ml correlate with the values for the 1st and 12th epochs, the estimates are biased due to the different method for calculating mean CoP position used to demean the CoP data prior to calculating the parameters (i.e., mean of CoP data within epoch vs. mean of CoP data for entire trial). Additionally, the correlations with whole-trial estimates largely disappear when comparing against transient characteristics that are defined by differences in epoch estimates. Collectively, these findings support that transient characteristics of postural sway may complement traditional whole-trial estimates.

Although transient behavior in CoP measures during quiet stance have previously been discussed, prior studies have not attempted to characterize aspects of the transient response using an epoch-based approach. Previous studies have identified transient effects in CoP measures following visual deprivation, but either had small sample sizes (3 participants) [9] or were restricted to clinical populations (diabetic neuropathy [8] and Parkinson's disease [10]). Some more recent studies have investigated relationships between transient postural behavior and sensory reweighting dynamics in healthy populations, albeit using more complex protocols and analysis techniques [20–22]. Additionally, numerous studies have generally viewed this initial behavior as undesirable noise with regard to its effects on obtaining traditional whole-trial estimates. Research aimed at maximizing the reliability of whole-trial estimates has understandably advocated for longer quiet stance trial durations [6, 23, 24]. In the context of our

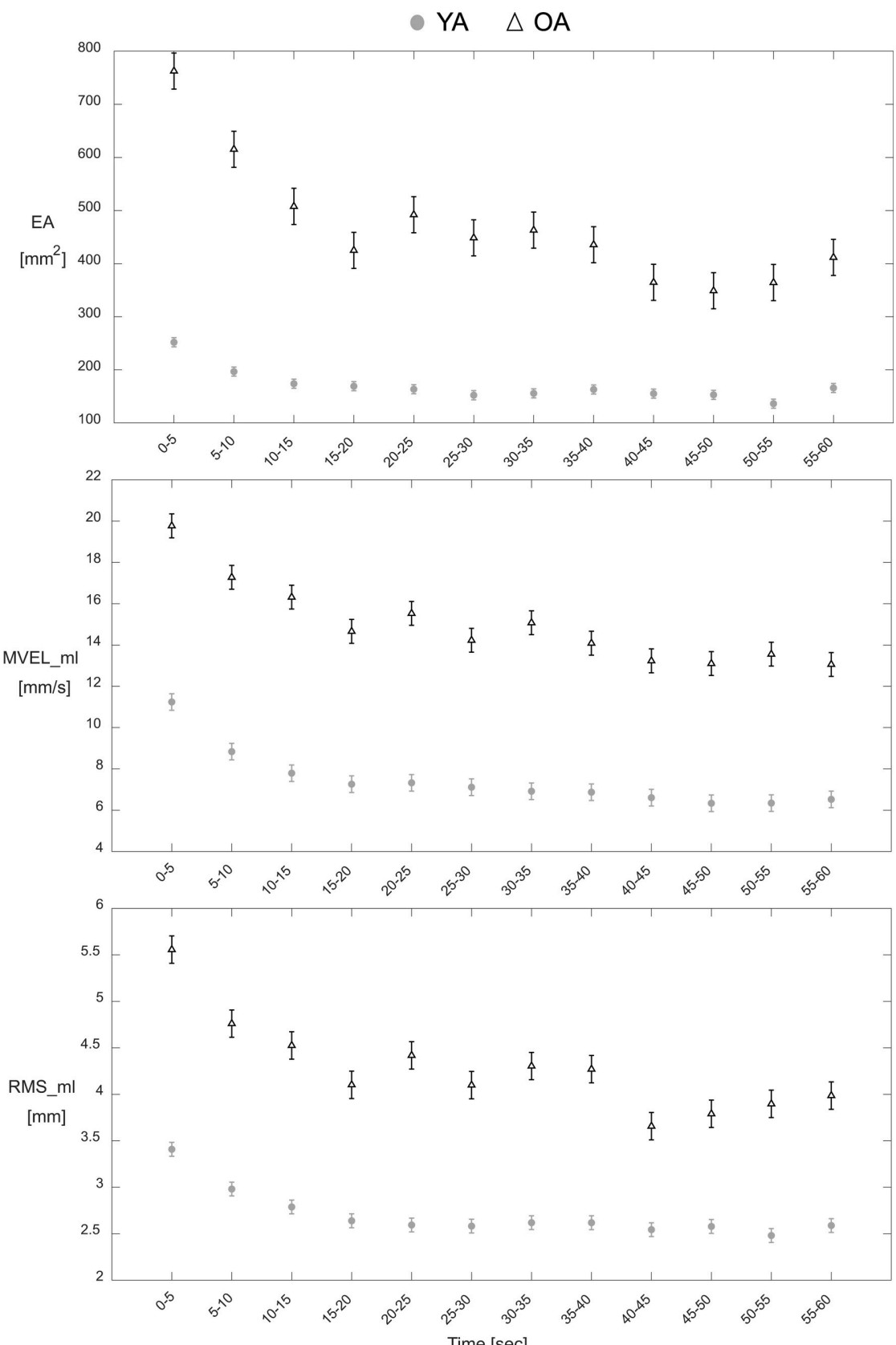

**Fig 3. Young and older adult transient behavior for each CoP parameter.** Gray circles and hollow triangles represent the time-series data for the Young Adult (YA) and Older Adult (OA) groups, respectively. Values correspond to mean ± standard error for each epoch.

findings, longer trials introduce greater proportions of a quasi-steady-state component (see 20–60 seconds region of Fig 2) that diminishes the influence of the initial transient portion (see 0–15 seconds region of Fig 2) on traditional whole-trial estimates. While our epoch-based approach gives insight into why longer trials are an effective strategy for increasing whole-trial estimate reliability, our findings also indicate that this strategy effectively marginalizes unique information that may be contained in this early portion of quiet stance trials. Our results indicate that particular consideration should be made regarding which aspects of postural control are most pertinent to a given hypothesis. Longer trials may improve the reliability of whole-trial estimates [6, 23, 24]; however, including a complementary assessment of the initial transient characteristics may provide a more comprehensive characterization of postural control by also accounting for potentially valuable information contained within initial transient responses. Further work is needed to determine if this more comprehensive approach provides advantages in terms of assessing deficiencies and predicting health outcomes.

Notably, we found significant differences between YA and OA groups in our study for the transient characteristics DIF_ovr for EA and DIF_ovr for RMS_ml. Where the OA group exhibited larger overall differences for EA and RMS_ml compared to the YA group. We speculate that the increased transient magnitude may be due to diminished sensory reweighting ability (i.e., the ability to respond to sensory conflicts or transitions) among the OA group. One potential explanation for this observation is that the OA group used a less effective strategy at the onset of the sensory transition of eye closure compared to the YA group. Previously, worse visual-somatosensory integration ability (i.e., the ability to consolidate sensory information from visual and somatosensory modalities) has been associated with worse balance and an increased likelihood of falling in older adults [25]. Although no study has investigated the direct relationship between sensory reweighting ability and fall risk [26], it may be beneficial to measure transient postural control responses to sensory transitions to understand how a person may respond to challenging real-world scenarios (e.g., lights turning off in a room). Additionally, it may be valuable to investigate if other sensory transitions (e.g. vestibular interference or somatosensation interference) elicit similar transient responses, as well.

Consistent with previous research that has established that older adults often exhibit diminished postural control and are at an elevated risk for falling compared to younger adults [3, 12, 13], these two transient characteristics (DIF_ovr for EA and DIF_ovr for RMS_ml) may also provide clinically-relevant information when assessing an individual's fall risk. While previous

**Table 5. Comparisons of transient characteristics between younger and older adult groups.**

| CoP Parameter | Transient Characteristic | YA | OA | P-value |
|---|---|---|---|---|
| **EA [mm²]** | DIF_ovr | 86.0 ± 133.0 | 350.9 ± 513.6 | **0.001**[*] |
| | DIF_13 | 78.0 ± 129.6 | 254.8 ± 514.9 | 0.064 |
| **MVEL_ml [mm/s]** | DIF_ovr | 4.72 ± 2.76 | 6.71 ± 7.45 | 0.704 |
| | DIF_13 | 3.45 ± 2.99 | 3.45 ± 8.35 | 0.246 |
| **RMS_ml [mm]** | DIF_ovr | 0.82 ± 1.03 | 1.57 ± 2.15 | **0.028**[*] |
| | DIF_13 | 0.62 ± 0.96 | 1.03 ± 2.09 | 0.496 |

[*] $P < 0.05$ from Kruskal Wallis tests

Mean ± SD values calculated for between-subjects.

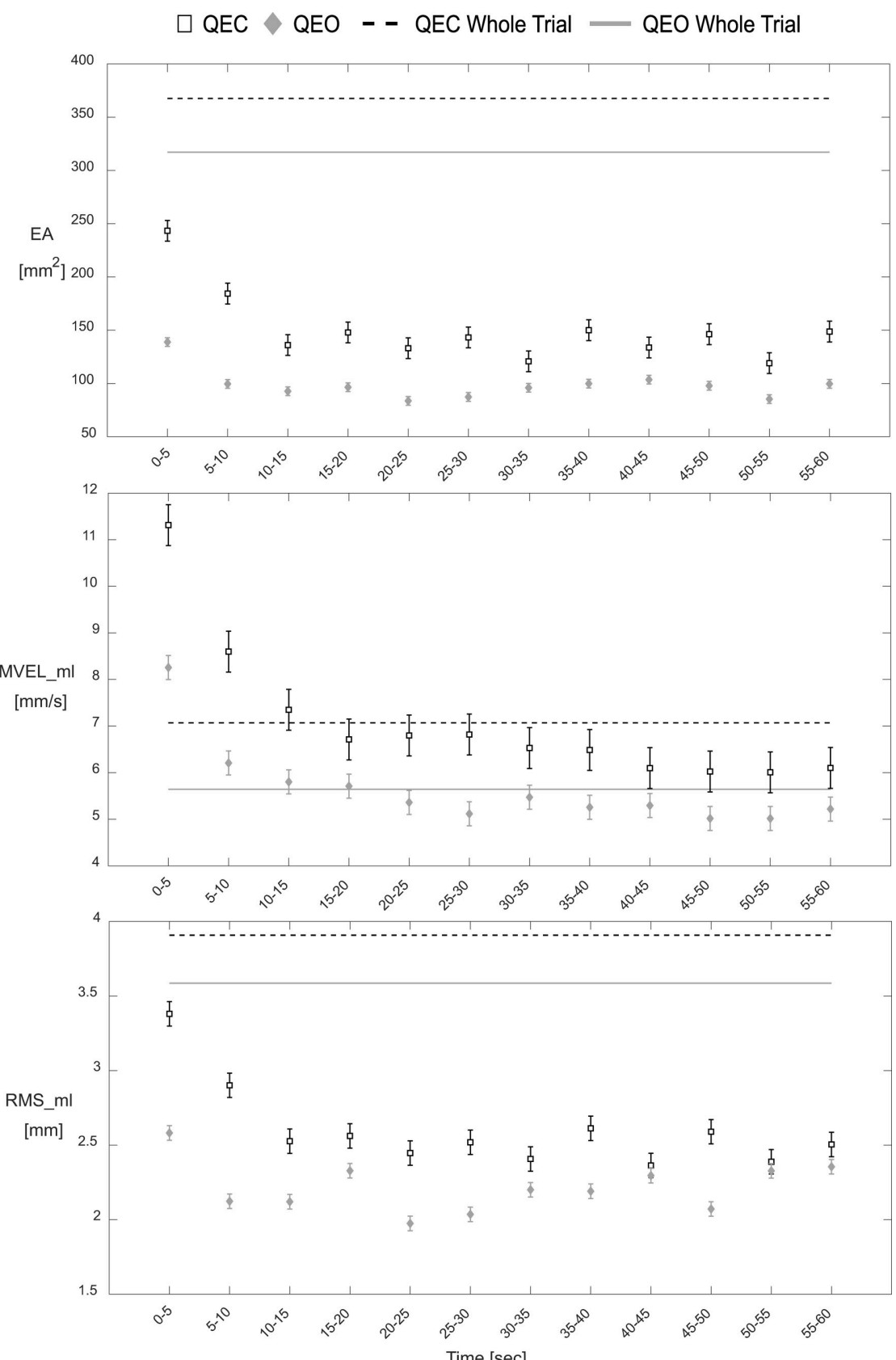

**Fig 4. Whole-trial estimates vs epoch-based estimates for each CoP parameter in QEC and QEO stance.** Hollow squares and gray diamonds represent the time-series data for QEC stance and QEO stance, respectively. Dashed black lines and solid gray lines represent the whole-trial estimates for QEC stance and QEO stance, respectively. Values correspond to mean ± standard error for each epoch.

research has identified numerous and sometimes contradictory CoP-based measures that are predictors of falls in older adults, there is no consensus on which CoP measures provide the best predictive ability for falls [4, 27, 28]. However, most of these studies used whole-trial CoP estimates and as established in this study, the transient characteristics of epoch-based CoP estimates exhibit negligible correlations with whole-trial estimates. Therefore, the transient characteristics DIF_ovr for EA and DIF_ovr for RMS_ml may offer complementary clinically-relevant information to be considered alongside whole-trial estimates for a more comprehensive assessment of postural control. Further studies that measure these transient characteristics and longitudinally track falls are necessary to assess whether transient characteristics combined with whole-trial estimates provide any improvement in the predictive ability for fall risk compared to traditional whole-trial estimates alone.

Interestingly, initial transient behavior consistently appeared during both QEC and QEO stance, although the effect on participants' balance was magnified in the eyes closed condition. These results were contrary to our original hypothesis that transient behavior would only exist during QEC stance as a response to the sensory transition that occurs when participants close their eyes and eliminate their vision from helping them control their balance. While it appears that sensory transitions do contribute to transient behavior, observing initial transient effects in the eyes open condition raises questions as to what other factors may be contributing to this behavior. Because participants were able to decide when each trial began by starting the '3-2-1-GO' countdown only after they felt comfortable and ready, we do not believe that the observed transient behavior is a result of participants adjusting to standing on a new surface or standing in the study-imposed stance. One potential contributing factor to the persisting transient effect during the eyes open condition may be a form of a cognitive perturbation that results from an individual transitioning from counting to standing still quietly. Future work is necessary to delineate additional factors that may contribute to transient behavior in postural control data.

As previously stated, during both QEC and QEO stance, we observed an initial transient period in which participants demonstrated worse balance compared to the rest of the trial. However, the transient behavior was exaggerated in the QEC condition (Fig 2), as defined by larger DIF_ovr and DIF_13 estimates compared to the QEO condition (Table 3). This result is consistent with previous work that reported transient responses associated with temporarily destabilizing effects following visual deprivation, although within diabetic neuropathy [8] and Parkinson's disease patients [10]. In addition, decreased postural stability following the withdrawal of vision has been reported, especially among older adults [29].

While this study provides novel insight into the transient characteristics of postural control, there are several limitations that should be considered. Due to the exploratory nature of this study, we did not correct for multiple comparisons. This provided a more stringent criteria for Experiment 1 where we are more likely to detect significant relationships, although still none were found. The risk of Type I error may be inflated in Experiments 2 and 3 and caution should be taken when considering the statistical significance of findings, particularly for comparisons with marginal $P$-values. However, even if we used an overly conservative Bonferroni correction [30, 31] for our 9 original dependent variables (0.05/9 = 0.0056), our results would still indicate that the DIF_ovr for MVEL_ml transient characteristic can distinguish between QEC and QEO stance, while DIF_ovr for EA can distinguish between YA and OA groups.

Future work may improve the ability for a DECAY coefficient approach to be used to gain insight into transient characteristics. While the exponential fit with an offset appropriately modeled the group-level epoch data, the quality of the fit for an isolated participant and condition (using average epoch values across trials for that person-condition) was much more variable. In some instances, no meaningful fits were able to be obtained. Therefore, it may be beneficial to obtain more than three trials for every participant and condition in order to account for the large variability in quality of fits for epoch data based only on three trials. Adjusting the epoch length may also alter the consistency of these fits. Improving the robustness of the fits would be particularly important for applying this method for screening or longitudinal tracking individual patients. Further work would also be needed to establish the reliability of the transient characteristics for tracking within-subject changes in postural control over time. Additionally, future work that investigates how various processing and filtering schemes affect the transient characteristics may be necessary.

Participants within the YA group from the Bozeman, MT community completed their testing in the Montana State University Neuromuscular Biomechanics Lab and wore noise-canceling headphones in order to ensure a quiet environment, free of audible and visual distractions. However, participants within the YA group from the 2017 ASB Quick Study and the OA group completed their testing at the 2017 ASB Annual Meeting and various senior living facilities, respectively. While steps were taken to minimize audible and visual distractions in these environments, they were not completely free of background noise due to the vibrant ASB conference being held in the same building. However, the protocols were otherwise identical and we do not believe that this introduced any confounding effects. Additionally, self-selection bias, particularly amongst the older adults that participated, may have resulted in a relatively higher performing OA group. By not having a truly representative sampling of a typical older adult population, the data may not be representative of the general older adult population. However, the large age difference between older and younger participants provides a useful starting place to understand the ability for features of transient behavior to distinguish between two groups with previously documented differences in postural control and fall risk [5].

## 6. Conclusion

This study provides insight into the transient behavior that is observed during quiet stance postural control in various age groups and under various sensory conditions. These findings indicate that using an epoch-based approach for analyzing postural control may capture unique and potentially clinically-relevant information that is marginalized when using traditional whole-trial estimates. Further evaluation is warranted to better understand the relationships between the observed transient behavior during quiet stance postural control and falls.

## Supporting information

**S1 Table. DECAY statistical analysis and results.**
(XLSX)

**S2 Table. 'Epoch' post-hoc analysis for all CoP parameters for eyes closed and eyes open stance.**
(XLSX)

**S3 Table. 'Trial Number' post-hoc analysis for all CoP parameters for eyes closed and eyes open stance.**
(XLSX)

**S4 Table. 'Epoch' post-hoc analysis for all CoP parameters for younger and older adult groups.**
(XLSX)

**S5 Table. 'Trial Number' post-hoc analysis for all CoP parameters for younger and older adult groups.**
(XLSX)

**S6 Table. Spearman's rank-order correlations between epochs 1 and 12 and whole-trial CoP estimates.**
(XLSX)

## Acknowledgments

We thank the Organizing Committee of the 2017 annual meeting of the American Society of Biomechanics (ASB), ASB 2017 attendees who participated, and the Ohio State University and University of Dayton volunteers for their support of the ASB Quick Study data collections. We would also like to thank the Montana State University Library for both the support of the Library Author Fund for open access charges associated with this publication and for the data publication in the Dryad repository. Finally, we would like to acknowledge the Montana State University Norm Asbjornson College of Engineering for their sponsoring the Benjamin Fellowship that supported Cody Reed during this work.

## Author Contributions

**Conceptualization:** Cody A. Reed, Ajit M. W. Chaudhari, Lise C. Worthen-Chaudhari, Kimberly E. Bigelow, Scott M. Monfort.

**Data curation:** Cody A. Reed, Scott M. Monfort.

**Formal analysis:** Cody A. Reed, Scott M. Monfort.

**Investigation:** Cody A. Reed, Ajit M. W. Chaudhari, Lise C. Worthen-Chaudhari, Kimberly E. Bigelow, Scott M. Monfort.

**Methodology:** Ajit M. W. Chaudhari, Lise C. Worthen-Chaudhari, Kimberly E. Bigelow, Scott M. Monfort.

**Project administration:** Ajit M. W. Chaudhari, Scott M. Monfort.

**Resources:** Ajit M. W. Chaudhari, Scott M. Monfort.

**Software:** Scott M. Monfort.

**Supervision:** Ajit M. W. Chaudhari, Scott M. Monfort.

**Visualization:** Cody A. Reed.

**Writing – original draft:** Cody A. Reed.

**Writing – review & editing:** Cody A. Reed, Ajit M. W. Chaudhari, Lise C. Worthen-Chaudhari, Kimberly E. Bigelow, Scott M. Monfort.

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
