## [Decision Letter · Decision Letter 0]

27 Apr 2020

PONE-D-20-04744

A new perspective on transient characteristics of quiet stance postural control

PLOS ONE

Dear Dr. Monfort,

Thank you for submitting your manuscript to PLOS ONE. After careful consideration, we feel that it has merit but does not fully meet PLOS ONE’s publication criteria as it currently stands. Therefore, we invite you to submit a revised version of the manuscript that addresses the points raised during the review process.

We would appreciate receiving your revised manuscript by Jun 11 2020 11:59PM. To enhance the reproducibility of your results, we recommend that if applicable you deposit your laboratory protocols in protocols.io, where a protocol can be assigned its own identifier (DOI) such that it can be cited independently in the future. For instructions see: http://journals.plos.org/plosone/s/submission-guidelines#loc-laboratory-protocols

We look forward to receiving your revised manuscript.

Kind regards,

Yumeng Li

Academic Editor

PLOS ONE

Journal Requirements:

Additional Editor Comments (if provided):

Some revisions and improvement are needed based on reviewers' comments.

Reviewers' comments:

Reviewer's Responses to Questions

**Comments to the Author**

1. Is the manuscript technically sound, and do the data support the conclusions?

Reviewer #1: Yes

Reviewer #2: Yes

2. Has the statistical analysis been performed appropriately and rigorously? 

Reviewer #1: Yes

Reviewer #2: Yes

3. Have the authors made all data underlying the findings in their manuscript fully available?

Reviewer #1: No

Reviewer #2: Yes

4. Is the manuscript presented in an intelligible fashion and written in standard English?

Reviewer #1: Yes

Reviewer #2: Yes

5. Review Comments to the Author

Reviewer #1: Reed et al. performed three different experiments to test if transient behavioral parameters yield different information compared to classical whole trial CoP parameters. By looking at different epoch-based characteristics, Reed et al could show that there is novel information in the transient behavioral parameters that might have a clinical impact in the future.

Major

1. Abstract is rather lengthy, too detailed and should be re-written and significantly shortened. I do like the first paragraph, the subsequent ones can be condensed to the central statement and the conclusion.

2. DECAY fitting. I do agree that the decay in an epoch-based setting is very interesting. However, I not agree with the function provided (ax^b), which is as far as I could see it not even written in the main text body. I suggest using a natural decay function (Ae^{-tx}) and determine t. The fits are clearly underestimating the data, as seen in Figure 2 and Figure 3. An exponential decay function will follow the trend more closely.

3. I would love to see in Figure 1 an overview over the three experiments (visualized protocol), as well as a workflow of the processing steps to determine the parameters used in the following. It helps readers of the manuscript a lot.

4. Figures: please remove the border of each subpanel and align the x-axis. This allows easy comparison of data trends

5. The data is not deposited anywhere, there’s only the sample text copied and pasted: “All XXX files are available from the XXX database (accession number XXX).”. Please provide an e.g. zenodo or figshare resource for the data.

Minor

- L106: What is the rational to use a 4th (!) order Butterworth filter?

- L131: How did the authors test for normal distribution?

- L137:  67

- L176, L233: comparisons were performed, but not made

- L314-316: I am not sure if I understand this correctly. If transient effects are averaged out means that there is no real effect, and one only observes noise at the beginning. It would be good to clarify this point.

- L368-369: Being not a statistics expert myself, I would still argue that you need to correct for multiple tests, regardless if your study is exploratory or not. As suggestion, you can use Bonferroni correction.

Reviewer #2: Manuscript # PONE-D-20-04744

Title: A new perspective on transient characteristics of quiet stance postural control

By CA Reed, AMW Chaudhari, LC Worthen-Chaudhari, KE Bigelow, SM Monfort.

General comments:

In this study, the authors provided insight into the transient behavior that is observed during quiet stance postural control in two age groups (young and old) and under two sensory conditions (i.e. eyes open and eyes closed). Their findings indicated that using an epoch-based approach for analyzing postural control could capture unique and potentially clinically-relevant information that could be difficult to capture when using traditional whole-trial estimates.

The authors designed and conducted this study reasonably well. I did not also find any concern about the analysis. So, I have only one minor comment on the discussion.

The idea of this study is so simple that it is hard to understand why previous studies did not assess stability of quiet stance postural control this way. It is helpful to add some description about a possible reason in the discussion.

6. PLOS authors have the option to publish the peer review history of their article (what does this mean?). If published, this will include your full peer review and any attached files.

Reviewer #1: No

Reviewer #2: Yes: Dr. Shinji Kakei

---

## [Author Response · Author response to Decision Letter 0]

18 Jun 2020

Reviewer Response for PLOSONE Submission PONE-D-20-04744 (This response letter is attached as a Word document and includes figures).

Introduction to Response to Reviewers: We greatly appreciate the efforts of the reviewers in assessing the merit of our manuscript for publication in PLOS ONE. We are encouraged by the reviewers’ interest in the manuscript and their feedback. We have taken their comments as opportunities to strengthen the manuscript and have revised the manuscript to address all reviewer comments. Specific reviewer comments are followed by responses from the co-authors, specific actions taken, and references to lines in the revised manuscript of actions taken, where applicable.

Reviewer 1 Comments for the Author...

R1.1

Abstract is rather lengthy, too detailed and should be re-written and significantly shortened. I do like the first paragraph, the subsequent ones can be condensed to the central statement and the conclusion.

----

Response: Thank you for the positive comments and constructive critiques that you have provided. In regard to this comment, we have shortened the abstract so that is approximately 25 words under the stated PLOS ONE guidelines for abstract word count. The shortened abstract provides a more concise and less-detailed outlook on the overall study and the 3 experiments contained within.

Action: Below is the shortened version of our abstract.

 “Postural control provides insight into health concerns such as fall risk but remains relatively untapped as a vital sign of health. One understudied aspect of postural control involves transient responses within center of pressure (CoP) data to events such as vision occlusion. Such responses are masked by common whole-trial analyses. We hypothesized that the transient behavior of postural control would yield unique and clinically-relevant information for quiet stance compared to traditionally calculated whole-trial CoP estimates. Three experiments were conducted to test different aspects of this central hypothesis.

To test whether transient, epoch-based characteristics of CoP estimates provide different information than traditional whole-trial estimates, we investigated correlations between these estimates for a population of young adults performing three 60-second trials of quiet stance with eyes closed. Next, to test if transient behavior is a result of sensory reweighting after eye closure, we compared transient characteristics between eyes closed and eyes open conditions. Finally, to test if there was an effect of age on transient behavior, we compared transient characteristics during eyes closed stance between populations of young and older adults.

 Negligible correlations were found between transient characteristics and whole-trial estimates (p>0.08), demonstrating limited overlap in information between them. Additionally, transient behavior was exaggerated during eyes closed stance relative to eyes open (p<0.044). Lastly, we found that transient characteristics were able to distinguish between younger and older adults, supporting their clinical relevance (p<0.029).

 An epoch-based approach captures unique and potentially clinically-relevant postural control information compared to whole-trial estimates. While longer trials may improve the reliability of whole-trial estimates, including a complementary assessment of the initial transient characteristics may provide a more comprehensive characterization of postural control.”

Reference: Lines 27-46

R1.2

DECAY fitting. I do agree that the decay in an epoch-based setting is very interesting. However, I not agree with the function provided (ax^b), which is as far as I could see it not even written in the main text body. I suggest using a natural decay function (Ae^{-tx}) and determine t. The fits are clearly underestimating the data, as seen in Figure 2 and Figure 3. An exponential decay function will follow the trend more closely.

----

Response: Thank you very much for the suggestion. This comment prompted a deeper look into the DECAY fitting analysis we used and provided us with a greater insight into our previous DECAY fits. In the preliminary stages of this work, we evaluated several options for curves of best fit for the epoch data including an exponential decay. This initial work led us to choose a power function (y = axb) because it provided the best fit out of commonly used curve-fitting functions. 

In response to this reviewer comment, we revisited the curves to determine whether this choice (power vs. exponential) was still appropriate. Similar to our initial exploration, we found that the natural decay (exponential) function to fit our data did not provide as adequate as a fit as the power function based on visual inspection and the correlation coefficient R2 (see Fig A below). However, we then attempted to characterize the transient behavior with an exponential function equation with a constant offset (y = a∙ebx + c). This function modeled the group data better than the original power function fit; however, we also identified that there was considerable variability in the appropriateness of the fit for individual trials and participants (average across 3 successful trials). Around 0-30% of the data for a given participant and condition were still not able to obtain a meaningful fit (R2 values were negative). Therefore, given the limitations of the poor fits, we elected to omit the DECAY analysis from the primary hypothesis testing for the three experiments, but have provided the associated analyses in the Supplemental content for reference to interested readers. 

Below we have provided a visual comparison of the power function, exponential decay, and exponential decay with a constant offset models for the group level EA data in Experiment 1 (Figure A). Additionally, we have provided a comparison of participant level EA data from Experiment 1 that demonstrate a good and a poor fit (Figure B). 

Figure A. Representative fits of group-level data. An exponential function with an added constant offset was the best fit equation.

Figure B. Example of large variability in the quality of the fit that was obtained from using the exponential function with a constant offset (which was also true for other equation options). Many trials displayed the characteristic transient decay followed by a quasi-steady-state period (i.e., ‘Good Fit’); however, some trials did not display any discernable patterns and were not modeled well by an exponential function (i.e., ‘Poor fit’). 

Action 1: We include the following paragraph in Section 2.1.2 that explains our attempt to fit the data with an exponential function with a constant offset.

“We also attempted to characterize the transient behavior with an exponential decay fit by using the exponent ‘b’, which we termed ‘DECAY’, from the exponential function equation with a constant offset ( y = a∙ebx + c ). While the group data (i.e., averaged epoch values across all trials and participants for a given condition or group) were modeled very well with the proposed exponential fit, considerable variability in the appropriateness of the fit was observed for fitting data for individual participants for a given condition. Poor fits were most prominent when fitting epoch data from individual trials, but persisted in 0-30% of participants when fitting epoch data that were averaged across trials for a given participant and condition. Cases of poor fits were highlighted by having negative R2 values. Given the limitations of the poor fits, we elected to omit the DECAY analysis from the primary hypothesis testing for the three experiments, but have provided the associated analyses in the Supplemental content for reference (S1 Table). In these supplemental analyses, we account for the poor fits using three approaches: 1) including all DECAY values as calculated, 2) omit the data where the fits do not provide meaningful information (i.e., negative R2), and 3) designate the DECAY values for these cases with a value of zero to indicate no observed transient behavior.”

Reference: Lines 120-131

Action 2: We omit all DECAY results and discussion from the manuscript and included these in the Supplemental content.

Reference: Fig 2,3,4. Table 1,2,3,5. Supplemental table S1.

Action 3: We include the following paragraph in the limitations portion of the Discussion section that proposes how a DECAY coefficient approach may be improved and used for future work when evaluating transient characteristics of postural control. 

“Future work may improve the ability for a DECAY coefficient approach to be used to gain insight into transient characteristics. While the exponential fit with an offset appropriately modeled the group-level epoch data (as seen in Fig 2,3), the quality of the fit for an isolated participant and condition (using average epoch values across trials for that person-condition) was much more variable. In some instances, no meaningful fits were able to be obtained. Therefore, it may be beneficial to obtain more than three trials for every participant and condition in order to account for the large variability in quality of fits for epoch data based only on three trials. Adjusting the epoch length may also alter the consistency of these fits. Improving the robustness of the fits would be particularly important for applying this method for screening or longitudinal tracking individual patients. Further work would also be needed to establish the reliability of the transient characteristics for tracking within-subject changes in postural control over time. Additionally, future work that investigates how various processing and filtering schemes affect the transient characteristics may be necessary.”

Reference: Lines 398-408

R1.3

I would love to see in Figure 1 an overview over the three experiments (visualized protocol), as well as a workflow of the processing steps to determine the parameters used in the following. It helps readers of the manuscript a lot.

----

Response/Action 1: We also believe that a visualization of an overview of the processing workflow for all three experiments could be greatly beneficial. Below is an updated version of Figure 1, with 1A representing the workflow overview and 1B the original Fig1 describing the transient characteristics visually.

Reference: Fig 1

Action 2: We have updated the Fig 1 caption to reflect the updated Fig 1.

Fig 1. Study overview. (A) General study workflow. Abbreviation definitions: YA = young adult, OA = older adult, QEC = quiet eyes closed, QEO = quiet eyes open. (B) Magnified visual representation of epoch-based estimates and transient characteristics DIF_ovr and DIF_13 for a hypothetical response of an example CoP parameter (e.g., EA, MVEL_ml, or RMS_ml).

Reference: Lines 133-136

R1.4

Figures: please remove the border of each subpanel and align the x-axis. This allows easy comparison of data trends

----

Action: The subpanels have been removed and the x-axes have been aligned for Figures 2, 3, and 4. Below is an example that shows the revised Figure 2. 

Reference: Fig 2, Fig 3, Fig 4

R1.5

The data is not deposited anywhere, there’s only the sample text copied and pasted: “All XXX files are available from the XXX database (accession number XXX).” Please provide an e.g. zenodo or figshare resource for the data.

----

Response: Thank you for the comment. We were operating on what we believed to be an option for the peer-review stage of the manuscript, but acknowledge the importance of ensuring the data underlying the results we report are adequately provided to improve reproducibility, transparency, and scientific rigor. 

Action: The data have now been deposited in the Dryad data repository under the title Data From: A new perspective on transient characteristics of quiet stance postural control. Currently, the submission is private and only available by accessing this link: https://datadryad.org/stash/share/Cmoos4w5D_pjV2ZVH56f9yH68FYi1YoCGyG7VqvEjh4.

When copied into a web browser, this link will download a zip file enclosing the spreadsheet containing all the underlying data and a readme text file that provides additional information about the dataset and clarifications on the abbreviations used in the submission. Upon acceptance of this manuscript, the data submission will become public through the Dryad data repository.

R1.6

L106: What is the rational to use a 4th (!) order Butterworth filter?

----

Response: The 4th order Butterworth filter has been used as a filtering strategy in previous studies investigating postural control, specifically References 5 (Prieto et al.) and 14 (Monfort et al.). As explained in David Winter’s seminal book Biomechanics and Motor Control of Human Movement, a 4th order filter provides a sharper cutoff frequency for the data. However, we do acknowledge that using a 4th order filter is not the only filtering strategy that has been used when assessing center of pressure data for postural control research. Therefore, we have added a sentence in the Discussion section that proposes the need to investigate how changing filtering schemes may affect the final results.

Action 1: We have added a citation after the sentence, “All data were 4th order Butterworth lowpass filtered at 20 Hz.” that includes the following references.

Prieto TE, Myklebust JB, Hoffmann RG, Lovett EG, Myklebust BM. Measures of postural steadiness: Differences between healthy young and elderly adults. IEEE Trans Biomed Eng. 1996;43: 956–966. doi:10.1109/10.532130

Monfort SM, Pan X, Patrick R, Singaravelu J, Loprinzi CL, Lustberg MB, et al. Natural history of postural instability in breast cancer patients treated with taxane-based chemotherapy: A pilot study. Gait Posture. 2016;48: 237–242. doi:10.1016/j.gaitpost.2016.06.011

Winter DA. Biomechanics and Motor Control of Human Movement: Fourth Edition. Biomechanics and Motor Control of Human Movement: Fourth Edition. 2009. doi:10.1002/9780470549148

Reference: Line 103

Action 2: We have added the following sentence when discussing the limitations of the study, “Additionally, future work that investigates how various processing and filtering schemes affect the transient characteristics may be necessary.”

Reference: Lines 406-408

R1.7

L131: How did the authors test for normal distribution?

----

Response: We tested for normal distribution using the Anderson-Darling Normality Test within Minitab.

Action 1: The following sentence in the statistical analysis section of Experiment 1 (Section 2.1.3) was modified to include a statement on how we tested for normal distribution, “Non-parametric tests were used because data were often not normally distributed, as assessed by the Anderson-Darling normality test.”

Reference: Lines 141-143

Action 2: The following sentence was added to the statistical analysis section of Experiments 2 and 3 (Sections 3.1.3 and 4.1.3), “Normality was assessed using the Anderson-Darling normality test.”

Reference: Lines 194-195 and 250-251

R1.8

L137:  67

----

Response: While we acknowledge that it is generally more concise to use numerals to represent numbers, we have adopted a consistent approach following recommendations of the American Medical Association (AMA) and American Psychological Association (APA) to spell out numbers that begin a sentence. With that said, if the PLOS ONE editorial staff prefer numerals at the start of the sentences, we are willing to adopt that approach.

R1.9

L176, L233: comparisons were performed, but not made

----

Action: The word “made” was replaced by “performed” in the lines mentioned above. These sentences were revised to read, “Tukey post-hoc comparisons were performed with a family error rate of α = 0.05.”

Reference: Lines 189 and 245

R1.10

L314-316: I am not sure if I understand this correctly. If transient effects are averaged out means that there is no real effect, and one only observes noise at the beginning. It would be good to clarify this point.

----

Response: We have attempted to clarify our point by adding in some additional details in this section that provide a more in-depth explanation of how trial duration impacts the contribution of transient effects to whole-trial CoP estimates.

Action: This paragraph of the discussion was revised to the following.

“Although transient behavior in CoP measures during quiet stance have previously been discussed, prior studies have not attempted to characterize aspects of the transient response using an epoch-based approach. Previous studies have identified transient effects in CoP measures following visual deprivation, but either had small sample sizes (3 participants) [9] or were restricted to clinical populations (diabetic neuropathy [8] and Parkinson’s disease [10]). Some more recent studies have investigated relationships between transient postural behavior and sensory reweighting dynamics in healthy populations, albeit using more complex protocols and analysis techniques [20–22]. Additionally, numerous studies have generally viewed this initial behavior as undesirable noise with regard to its effects on obtaining traditional whole-trial estimates. Research aimed at maximizing the reliability of whole-trial estimates has understandably advocated for longer quiet stance trial durations [6,23,24] . In the context of our findings, longer trials introduce greater proportions of a quasi-steady-state component (see 20-60 seconds region of Fig 2) that diminishes the influence of the initial transient portion (see 0-15 seconds region of Fig 2) on traditional whole-trial estimates. While our epoch-based approach gives insight into why longer trials are an effective strategy for increasing whole-trial estimate reliability, our findings also indicate that this strategy effectively marginalizes unique information that may be contained in this early portion of quiet stance trials. Our results indicate that particular consideration should be made regarding which aspects of postural control are most pertinent to a given hypothesis. Longer trials may improve the reliability of whole-trial estimates [6,23,24]; however, including a complementary assessment of the initial transient characteristics may provide a more comprehensive characterization of postural control by also accounting for potentially valuable information contained within initial transient responses. Further work is needed to determine if this more comprehensive approach provides advantages in terms of assessing deficiencies and predicting health outcomes.”

Reference: Lines 328-346

R1.11

L368-369: Being not a statistics expert myself, I would still argue that you need to correct for multiple tests, regardless if your study is exploratory or not. As suggestion, you can use Bonferroni correction.

----

Response: While we acknowledge that we have not corrected for multiple comparisons in our study, there is support within statistical literature for this approach, especially for exploratory studies (see references below). By correcting for multiple comparisons and reducing the likelihood of Type I error, the chance of making Type II errors increases. Thus, the main argument against correcting for multiple comparisons for exploratory studies, is that in doing so, potentially meaningful findings to guide future confirmatory research may be missed (i.e., found non-significant). While we understand the rationale for adjusting for multiple comparisons, we believe that our approach is appropriate for the present study. In particular, by not correcting for the multiple comparisons in Experiment 1 to test our hypothesis that the epoch-based approach characterizes different balance features than the whole-trial approach, we are adopting a more strict approach (i.e., we are more likely to identify a ‘significant’ finding that would counter our hypothesis). Additionally, it is noteworthy that correcting for multiple comparisons in Experiments 2 and 3 using an overly conservative Bonferroni correction for our 9 dependent variables (0.05/9=0.0056) would still yield the same conclusions (although notably some differences in what variables reached statistical significance). Therefore, we believe the approach we used is most appropriate for this particular study in order to provide guidance for future work in the area of the novel approach we present. We have added additional discussion regarding limitations associated with not correcting for multiple comparisons and the minimal impact we believe it imposes on the conclusions of our manuscript. 

Action 1: We have added the following citations to the discussion on why we chose not to correct for multiple comparisons.

Rothman KJ. No adjustments are needed for multiple comparisons. Epidemiology. 1990;1: 43–46. doi:10.1097/00001648-199001000-00010

Perneger T V. What’s wrong with Bonferroni adjustments. Br Med J. 1998;316: 1236–1238. doi:10.1136/bmj.316.7139.1236

Reference: Line 395

Action 2: We have added the following sentences when discussing the limitations of not correcting for multiple comparisons, “Due to the exploratory nature of this study, we did not correct for multiple comparisons. This provided a more stringent criteria for Experiment 1 where we are more likely to detect significant relationships, although still none were found. The risk of Type I error may be inflated in Experiments 2 and 3 and caution should be taken when considering the statistical significance of findings, particularly for comparisons with marginal P¬-values. However, even if we used an overly conservative Bonferroni correction [30,31] for our 9 original dependent variables (0.05/9 = 0.0056), our results would still indicate that the DIF_ovr for MVEL_ml transient characteristic can distinguish between QEC and QEO stance, while DIF_ovr for EA can distinguish between YA and OA groups.”

Reference: Lines 391-397

R2.1

Reviewer 2 Comments for the Author...

The idea of this study is so simple that it is hard to understand why previous studies did not assess stability of quiet stance postural control this way. It is helpful to add some description about a possible reason in the discussion.

----

Response: We appreciate the reviewer’s support for our work. In responding to Reviewer 1 (R1.10), we have added additional discussion regarding this point. Notably, several researchers have reported transient behavior in quiet stance CoP measures, however initial transient behavior has generally been viewed as ‘noise’ when the primary interest is in obtaining reliable whole-trial estimates. Our perspective contrasts this viewpoint by considering the transient behavior as a feature of postural control, which is more consistent with a sensory reweighting paradigm. Notably, we hope the simplicity in the approach may reduce barriers for researchers to complement traditional whole trial estimates with measures of transient behavior, which will advance our understanding of their clinical utility.

Action: We have revised the Discussion to include the following paragraph that discusses this content in more detail. Please refer to response/action to comment R1.10 for additional details

Reference: Lines 328-346

---

## [Decision Letter · Decision Letter 1]

23 Jul 2020

A new perspective on transient characteristics of quiet stance postural control

PONE-D-20-04744R1

Dear Dr. Monfort,

We’re pleased to inform you that your manuscript has been judged scientifically suitable for publication and will be formally accepted for publication once it meets all outstanding technical requirements.

Kind regards,

Yumeng Li

Academic Editor

PLOS ONE

Additional Editor Comments (optional):

The authors have successfully addressed all comments of the reviewers. The paper is accepted in its current form.

Reviewers' comments:

Reviewer's Responses to Questions

**Comments to the Author**

1. If the authors have adequately addressed your comments raised in a previous round of review and you feel that this manuscript is now acceptable for publication, you may indicate that here to bypass the “Comments to the Author” section, enter your conflict of interest statement in the “Confidential to Editor” section, and submit your "Accept" recommendation.

Reviewer #1: All comments have been addressed

Reviewer #2: (No Response)

2. Is the manuscript technically sound, and do the data support the conclusions?

Reviewer #1: Yes

Reviewer #2: (No Response)

3. Has the statistical analysis been performed appropriately and rigorously? 

Reviewer #1: Yes

Reviewer #2: (No Response)

4. Have the authors made all data underlying the findings in their manuscript fully available?

Reviewer #1: Yes

Reviewer #2: (No Response)

5. Is the manuscript presented in an intelligible fashion and written in standard English?

Reviewer #1: Yes

Reviewer #2: (No Response)

6. Review Comments to the Author

Reviewer #1: The authors did a great job in answering my comments and tackling my concerns/suggestions. I reviewed the changes and the deposited data, and agree with the authors to remove the DECAY part of the main manuscript and leave it open for further analysis/studies. I recommend this paper to be accepted in its current form.

Reviewer #2: (No Response)

7. PLOS authors have the option to publish the peer review history of their article (what does this mean?). If published, this will include your full peer review and any attached files.

Reviewer #1: No

Reviewer #2: No

---

## [Editor Report · Acceptance letter]

30 Jul 2020

PONE-D-20-04744R1 

A new perspective on transient characteristics of quiet stance postural control 

Dear Dr. Monfort:

I'm pleased to inform you that your manuscript has been deemed suitable for publication in PLOS ONE. Congratulations! Your manuscript is now with our production department. 

Kind regards, 

on behalf of

Dr. Yumeng Li 

Academic Editor

PLOS ONE